# Towards Complex Tissues Replication: Multilayer Scaffold Integrating Biomimetic Nanohydroxyapatite/Chitosan Composites

**DOI:** 10.3390/bioengineering11050471

**Published:** 2024-05-09

**Authors:** Barbara Palazzo, Stefania Scialla, Amilcare Barca, Laura Sercia, Daniela Izzo, Francesca Gervaso, Francesca Scalera

**Affiliations:** 1ENEA, Division for Sustainable Materials, Brindisi Research Center, S.S. 7 Appia Km. 706, 72100 Brindisi, Italy; barbara.palazzo@enea.it; 2Institute of Polymers, Composites and Biomaterials, National Research Council (IPCB-CNR), 80125 Naples, Italy; stefania.scialla@cnr.it; 3Laboratory of Applied Physiology, Department of Experimental Medicine, University of Salento, Campus Ecotekne, 73100 Lecce, Italy; amilcare.barca@unisalento.it; 4Centre for Regenerative Medicine “Stefano Ferrari”, Department of Life Sciences, University of Modena and Reggio Emilia, 41125 Modena, Italy; laura.sercia@unimore.it; 5Department of Engineering for Innovation, University of Salento, 73100 Lecce, Italy; 6CNR NANOTEC—Institute of Nanotechnology, Campus Ecotekne, Via Monteroni, 73100 Lecce, Italy

**Keywords:** chitosan, hydroxyapatite, biomineralization, multilayered scaffolds

## Abstract

This study explores an approach to design and prepare a multilayer scaffold mimicking interstratified natural tissue. This multilayer construct, composed of chitosan matrices with graded nanohydroxyapatite concentrations, was achieved through an in situ biomineralization process applied to individual layers. Three distinct precursor concentrations were considered, resulting in 10, 20, and 30 wt% nanohydroxyapatite content in each layer. The resulting chitosan/nanohydroxyapatite (Cs/n-HAp) scaffolds, created via freeze-drying, exhibited nanohydroxyapatite nucleation, homogeneous distribution, improved mechanical properties, and good cytocompatibility. The cytocompatibility analysis revealed that the Cs/n-HAp layers presented cell proliferation similar to the control in pure Cs for the samples with 10% n-HAp, indicating good cytocompatibility at this concentration, while no induction of apoptotic death pathways was demonstrated up to a 20 wt% n-Hap concentration. Successful multilayer assembly of Cs and Cs/n-HAp layers highlighted that the proposed approach represents a promising strategy for mimicking multifaceted tissues, such as osteochondral ones.

## 1. Introduction

Multilayered tissues, such as skin, vessels, and so on, have complex structures with physical and chemical gradients. Particularly, mineralized tissues, i.e., musculoskeletal and dental tissues, frequently comprise a heterogeneous collection of tissues working in unison. For example, the periodontal complex and the osteochondral unit are composed of multiple interfacing tissue types. The periodontium is tooth-supporting tissue created by gingiva, periodontal ligament, cementum, and alveolar bone. On the other hand, the osteochondral unit is composed of two closely interconnected but very dissimilar tissues, the articular cartilage and the underlying subchondral bone.

In this paper, osseocartilaginous tissue has been considered as a structural model of a typical multilayered tissue.

This is a non-vascularized and poorly cellularized multifaceted tissue that is frequently damaged as a result of important trauma or repetitive minor trauma. The damage includes pathological changes in the cartilage and in the underlying bone. Moreover, untreated lesions, evolving into irreversible joint damage, often lead to osteoarthritis [1,2,3,4]. Considering that cartilage is a tissue with poor reparative potential [5], which progressively decreases with age, the treatment of osteochondral lesions is complex.

Depending, for example, on the defect size (usually within 2 cm^2^ or larger), microfracturing, a minimally invasive arthroscopic technique that induces the migration of mesenchymal stem cells from bone marrow to the site of a cartilage defect [6], or mosaicplasty surgery using autografts or scaffolds [7] is necessary.

Osteochondral allograft use is limited by graft availability and, moreover, failure of cartilage incorporation is an issue [8]. Thus, replicating the spatial complexity of osteochondral units in terms of composition, structure, and function has not yet been accomplished. Despite considerable successes in the field of regenerative medicine that have allowed for achieving predictable and durable repair after an articular cartilage knee injury, obtaining physiological chondral tissue and not fibrocartilage is still a clinical challenge for orthopedic surgeons. [9]

To reach this aim, both cartilage and bone tissues need to be mimicked through 3D scaffolds. Osteochondral substitute architectures have experienced a huge improvement from the simplest monophasic scaffolds [10,11,12,13,14,15], to biphasic [16,17,18], triphasic [19,20], multiphasic [21,22,23], and most recently, gradient ones [24,25,26] to better mimic the complex hierarchy of native osteochondral units.

A single-phase scaffold contains a uniform material composition and structure, and despite its simplicity in processing, its structural design, i.e., macroscopic shape, pore size, pore topology, and orientation, can induce effective tissue renewal in specific applications.

However, the osteochondral unit is composed of two closely interconnected but very dissimilar tissues, the articular cartilage and the underlying subchondral bone. As a consequence, an efficient osteochondral tissue substitute should be able to mimic both biological materials in their structure and functionality, which led to the necessity of designing biphasic scaffolds with two different regions resembling the stratified anatomical architecture. Biphasic scaffolds should also guarantee strict integration between the organic and the inorganic parts, mimicking the biological tissue between the chondral and the osseous phases [27]. Moreover, to also mimic the intermediated zone, through which the articular cartilage is connected to the subchondral bone, i.e., the calcified cartilage, triphasic scaffolds [19,20] can be envisioned.

In addition, since native osteochondral tissue possesses a much more complicated gradient heterogeneity instead of a stratification of two or three separate regions, multiphasic discrete [21,22,23] or continued gradient [24,25,26] scaffolds have been proposed to achieve gradients throughout the entire construct or within a limited interfacial region.

Despite the above-highlighted improvements proposed in the field, the clinical products for cartilage and osteochondral defect repair currently available on the market are very limited, as reported in a recent review by Wei and Dai [28] on articular cartilage and osteochondral tissue engineering techniques.

Following this discussion in a top–down manner, we can consider that, if the above-cited methods can allow resembling the osteochondral macroscopic hierarchy, the implementation of a biomimetic path may introduce the physiological hierarchy present at a nano-dimensional level, allowing approaching the natural osteochondral tissue structure at a multiscale level, from macro to nano. In fact, in nature, the assembly of biological molecules or macromolecules and, in the case of bone, inorganic counterparts, i.e., the hybridization process, occurs at the nanoscale. In this way, nature takes advantage of the higher strength and flaw tolerance of nanoscale inorganic building blocks to enhance the mechanical properties of a wide range of biological composites. For this reason, bone is the typical “product” of an organic matrix-mediated mineralization process. Therefore, based on these considerations, mimicking at least some of the aspects of biological mineralization can be an effective strategy to assemble bone-like hydroxyapatite (HAp) that is close to natural bone with low crystallinity and nanoscale size, and to produce a composite with an intimate interaction between the organic and inorganic counterparts.

Researchers have been using different mineralization methods based on polymers as organic templates to regulate the nucleation and growth of calcium phosphate minerals (CaP), including the wet chemical method, the simulated body fluid or artificial saliva soaking method, and the alkaline phosphatase enzyme-induced method [29,30]. Polymers indicated for biomineralization contain loci for calcium ion sorption and consequent phosphate attraction, i.e., carboxylate or amino or hydroxyl moieties. Among natural polymers, ideal candidates include, for example, collagen, sometimes in association with polyaspartate, alginate, hyaluronic acid, starch-derived polymers, and finally chitosan [31,32]. Chitosan is a naturally abundant, cost-effective, and biodegradable polysaccharide that exhibits polyelectrolyte action and represents a suitable organic template, which, combined with biomimetic mineralization technology, can be used to develop organic-inorganic composite materials for hard tissue repair. Despite the above-cited methods being suitable for growing dentine or bone-like CaP crystals onto scaffold surfaces, they can take time (SBF), not be cost-effective (ALP), and not ever be combinable with the needs of a scaffold that must be suited to an osteochondral defect. In this paper, to the best of our knowledge, a chitosan/nanohydroxyapatite (Cs/n-HAp) composite scaffolds have been realized for the first time by combining the freeze-drying technique with an “in situ” biomineralization process. Our rationale was to join the possibility of tailoring the scaffold porosity and shape using the freeze-drying technique on well-developed gels with the opportunity of growing bone-like minerals, controlling their crystallinity, dimension, and percentage with respect to the polymer. Following this logic, the design of a graded composite Cs-nucleated HAp scaffold, which mimics the chemical-physical properties of the different layers of native osteochondral structures, is proposed. This scaffold was prepared by assembling n-HAp/Cs composite porous membranes with different weight percentages of inorganic phase (0–30 wt%), which in turn were obtained by growing crystals into single layer structures. Definitely, the aim of this work is to develop an innovative method for nucleating a bone-like n-HAp into the Cs matrix (which has been optimized from our previous work) [33] and its combination with an original stacking technique. The combination of these approaches allows for the design of a multilayer substitute at increasing concentrations of HAp along the z-axis, the configuration of which can be tailored as a function of the multilayered unit to be mimicked.

## 2. Materials and Methods

### 2.1. Chemicals

Low molecular weight chitosan (Cs, Mw 50–190 kDa based on viscosity, degree of deacetylation ≥ 75%) was purchased from Fluka. Acetic acid (ACS reagent ≥ 99.7%), calcium acetate hydrate (Ca(ac)_2_ H_2_O), phosphoric acid (H_3_PO_4_ ≥ 85 wt% in water), L-arginine (Arg, ≥ 98 wt%), phosphate-buffered saline (PBS) tablets, 2-ammino-2-(idrossimetil)-1,3-propandiolo (Trizma^®^ base), sodium chloride (NaCl ≥ 99.0 wt%), and sodium azide (NaN_3_ ≥ 99.0 wt.%) were from Sigma-Aldrich (Darmstadt, Germany). All chemicals were used as received. Ultrapure water (ρ > 18.2 MΩ·cm at 25 °C) was used for all experiments.

### 2.2. Cs/n-HAp Scaffold Preparation

Cs/n-HAp scaffolds loaded with different weight percentages of inorganic phase were obtained by combining a freeze-drying technique with an “in situ” nucleation method triggered by a basic amino acid, L-arginine (Arg), as partially described in our previous studies [34]. The methodology presented herein consists of three main phases, as illustrated in Figure 1A: (i) gel preparation; (ii) freeze-drying plus solvent thermal removal; and (iii) n-HAp “in situ” nucleation. (i) Gel preparation: A Cs-based gel (final concentration 1.67% *w*/*v*), containing n-HAp precursors was prepared by dissolving Cs powder into acetic acid solution. Specifically, Ca(ac)_2_ and H_3_PO_4_ solutions were added to the Cs gel (900 mg Cs dissolved in 36 mL acetic acid aqueous solution, at 1, 2, and 3% *w*/*v*, respectively) and mixed until reaching homogeneous suspensions. Three precursor concentrations were used to serve a Ca/P molar ratio equal to 1.67 and in order to theoretically obtain final n-HAp loadings of 10, 20, and 30 wt% with respect to the composite. Cs and Cs/precursor compositions are listed in Table 1. (ii) Freeze-drying plus solvent thermal removal: Air bubble-free slurries were then poured into 6 cm petri dishes and underwent a freeze-drying program (SP VirTis AdVantage Pro Freeze Dryer, SP Industries, Inc., Warminster, UK), involving a freezing step at −40 °C (freezing rate 1 °C/min) followed by a drying phase at 0 °C under vacuum for 19 h, as already detailed [34]. A post-processing treatment aimed to decrease the acetic acid concentration within the freeze-dried membrane layers: pure Cs and Cs/precursor scaffolds were thermally treated in a vacuum stove (Vuotomatic 50—BICASA, Milan, Italy) at 100 °C, p = −1 bar, for 3 h. Afterwards, cylindrical scaffolds (d = 6 mm, h = 4 mm) were obtained by cutting the freeze-dried membrane using a biopsy punch. (iii) n-HAp “in situ” nucleation: Pure Cs and Cs/precursor-loaded (10, 20 and 30 wt%) cylindrical scaffolds were incubated in a 0.05 M L-Arg solution under vacuum at 37 °C for 24 h to induce n-HAp nucleation inside the highly porous Cs scaffolds consequently to a pH increase. The obtained composites were finally washed in milli-Q water until neutral pH was reached and then freeze-dried again to remove the excess solvent.

### 2.3. Cs/n-HAp Multi-Layer Scaffold Prototype Preparation: Proof-of-Concept

For the multilayer fabrication, each suspension (Cs, Cs/precursor) was poured into a properly designed PLA mold, realized by using a 3DPRN LAB 3D (TIPS, Castiglione M.R. (TE), Italy). These layers, frozen separately, were then glued to each other by a suitable volume of chitosan suspension depending on the area of the surfaces to adhere (e.g., about 200 µL was sufficient for adhesion of two membranes with a surface area of 230 mm^2^). The multilayer structure thus obtained was frozen at −40 °C (freezing rate of 1 °C/min) to guarantee perfect adhesion between the layers, and subsequently freeze-dried for 24 h. The following solvent removal and n-HAp nucleation phases were executed in the same manner as for the single-layer scaffolds. The layers’ stacking mode will be shown later in the discussion.

### 2.4. Scaffold Physicochemical Characterization

#### 2.4.1. Morphological and Compositional Analysis (SEM-EDS)

The inner porous structure of the freeze-dried scaffolds, in terms of spatial distribution, size, shape, and degree of interconnection of the pores, was investigated using a scanning electron microscope (SEM, Zeiss Evo 40, Jena, Germany). Cross and longitudinal sections of the scaffolds coated with gold (*n* = 3 scaffolds for each type) were acquired at three different magnifications (60×, 150×, and 300×) at 20 keV of voltage. SEM micrographs were then processed and analyzed with ImageJ 1.50c. software (NIH, http://rsb.info.nih.gov/ij accessed on 10 January 2024) to determine the average pore diameter.

The scaffold porosity was calculated using the following Formula (1):(1)$$\mathrm{Porosity}\;(\%)=\left(1-\frac{\rho^{\prime }}{\rho }\right)\times 100$$
where $\rho^{\prime }$ refers to the density of the scaffold, calculated as the ratio of mass and volume, and ρ is the intrinsic density of the composite material, calculated considering the percentages of chitosan and the mixed precursors [13,35].

The SEM observations were matched with energy dispersive X-ray spectroscopy (EDS) using an XFlash Detector 5010 (Bruker, Berlin, German). Specifically, the calcium (Ca) and the phosphorous (P) distributions and relative mapping within the porous scaffolds (*n* = 3 scaffolds for each type) were evaluated considering three regions for each sample at 100× magnification using Quantax 400 Esprit 1.8.5 software.

#### 2.4.2. Structural Analysis by XRD

The scaffolds were examined by X-ray diffraction analysis (XRD) using a D-Max/Ultima X-ray diffractometer (Rigaku, Tokyo, Japan) operated at 40 kV and 40 mA, using Cu Kα radiation (λ = 1.54 Å) at a step size of 0.02°/step, and a scanning speed of 3 s/step. The average crystallite domain size of the nucleated n-HA (along the [002] and [310] directions for n-HA) was estimated applying Scherrer’s Equation (2):(2)$$D=\frac{k\lambda }{\beta cos\theta }$$
where k is a dimensionless crystallite shape factor; λ is the X-ray wavelength (Cu Kα1); β is the line broadening at half the maximum intensity (FWHM), after subtracting the instrumental line broadening (in radians); and θ is the Bragg’s diffraction angle for a plane (hkl).

#### 2.4.3. FTIR

The infrared spectra were recorded in the wavenumber range from 4000 to 400 cm^−1^ with a spectral resolution of 2 cm^−1^ using a JASCO FT/IR-6300 spectrometer empowered by Spectra Manager Version 2 software (JASCO Europe srl, Cremella, LC, Italy). Powdered samples (approximately 1 mg) were mixed with about 199 mg of anhydrous potassium bromide (KBr, 99.9% from Sigma-Aldrich). The mixture was pressed into 7 mm diameter tablets. A pure KBr disk was used as a blank, and the spectrum ordinates were converted from transmittance into absorbance units.

#### 2.4.4. Thermogravimetric Analysis

Thermogravimetric analysis (TGA) investigations were carried out on dried samples using a TA Instruments SDTQ600 Analyzer (New Castle, DE, USA). Heating was performed in a nitrogen flow (50 mL·min^−1^) using an alumina sample holder at a rate of 10 °C·min^−1^ up to 1200 °C. The inorganic content in the nanocomposite was determined by the residue in TGA through the following Equation (3):(3)$$nHAp\left(\%\right)=\left(composite\;residue\;\%\right)-(Cs\;residue\;\%)$$
in which the composite residue % and Cs residue % are the solid residues obtained by heating the composite and crosslinked pure chitosan scaffolds at 1200 °C, respectively.

#### 2.4.5. Swelling Study

The swelling behavior of the Cs and Cs/n-Hap scaffolds was assessed by evaluating the weight variation of the scaffolds over time. Freeze-dried scaffolds were weighed (w_d_), incubated in 1× PBS, pH 7.4, at 37 °C, and weighed (w_w_) in the hydrated form at scheduled times, up to 72 h. After weighing, samples were immersed in fresh buffer solution in order to continue the swelling kinetics. The swelling ratio percentage (Q, %) was calculated using the following Equation (4), where w_d_ is the initial dry weight of the sample and w_w_ is the weight of the swollen scaffolds at each time point:(4)$$Q\;(\%)=\frac{w_{w}-w_{d}}{w_{w}}\cdot 100$$

#### 2.4.6. Mechanical Properties

The mechanical properties of the Cs and Cs/n-Hap scaffolds were investigated by means of uniaxial unconfined stress–strain compression tests. Cylindrical scaffolds (Ø = 6 mm, h = 4 mm) were preliminarily hydrated in PBS overnight at 4 °C, and the diameter and thickness of the swollen scaffolds were recorded before starting the test. Each scaffold was then mounted on a Z1.0 TH testing machine (Zwick Roell, Germany), equipped with a 10 N load cell. The swollen scaffolds were compressed at room temperature under displacement control up to 75% of strain (rate = 0.01 mm·s^−1^, pre-load = 0.01 N), while immersed in PBS. The non-linear mechanical behavior of the scaffolds was assessed by estimating the following three parameters extrapolated from the compressive stress–strain curve: the Young modulus at both low and high strain values and the strain value at which the stress–strain curve drastically changed its slope. The Young’s modulus was calculated as the slope of the linear part of the stress–strain curve between 2% and 10% (E_low_) and 70–75% (E_high_), and the strain at which the stress–strain curve stiffened was calculated as the x-axis intercept of the E_high_ straight line (ε_y_ = 0).

### 2.5. Preliminary In Vitro Biological Validation

#### 2.5.1. In Vitro Cell Maintenance

Cytocompatibility studies of the Cs/n-HAp scaffolds were performed using the mouse embryonic fibroblast cell line (NIH-3T3, ATCC CTRL-1658™) as an in vitro model. NIH-3T3 cells were cultured in sterile T75 cm^2^ cell culture flasks in Dulbecco’s Modified Eagle Medium low glucose (DMEM; Sigma-Aldrich, Milan, Italy) supplemented with 10% (*v*/*v*) fetal serum bovine (FBS, Sigma-Aldrich, Milan, Italy), 100 μg/mL penicillin-streptomycin solution (Pen/Strep, Sigma-Aldrich, Milan, Italy), and 2 mM of L-glutamine (L-Glu, Sigma-Aldrich, Milan, Italy) in a water-saturated atmosphere of 5% *v*/*v* CO_2_ and 95% *v*/*v* air at 37 °C. Adherent cells were washed twice with Dulbecco’s Phosphate-Buffered Saline (D-PBS, Sigma-Aldrich, Milan, Italy), and then detached from the flask’s surface by adding D-PBS containing 2 mM trypsin-EGTA (Sigma-Aldrich, Milan, Italy) for 1–5 min at 37 °C, once reaching 70–90% confluence (every 2–3 days). Trypsin action was blocked by adding a suitable volume of culture medium (containing α2-antitrypsin). Cell suspensions were harvested by centrifugation (1200 rpm for 7 min, Beckman Coulter Allegra 6R centrifuge with the GH-3.8 swing bucket rotor). The obtained pellets were re-suspended in fresh culture medium and transferred to new flasks. For all experiments, cultures between passages 3 and 7 of propagation were used.

#### 2.5.2. Cell Viability Assay

Before cell seeding, the Cs and Cs/n-HAp cylinder scaffolds (d = 6 mm, h = 4 mm) were sterilized by UV irradiation for 2 h. Then, the scaffolds were placed in a 96-well plate and hydrated under sterile conditions in a water-saturated atmosphere of 5% *v*/*v* CO_2_ at 37 °C for 24 h. NIH-3T3 cells were seeded dropwise onto the top of the scaffold (1·10^5^ cells/scaffold in 100 μL of medium) and incubated in a water-saturated atmosphere of 5% *v*/*v* CO_2_ at 37 °C for 1 h. Next, additional fresh medium was added dropwise (100 μL/well) to the scaffolds and the cells were maintained in culture for 48 h. NIH-3T3 fibroblast cell proliferation on the Cs and Cs/n-HAp scaffolds was determined via the 3-(4,5-dimethylthiazol-2-yl)-2,5-diphenyltrazolium bromide (MTT) assay, a method based on the metabolic activity of viable cells to convert MTT into an insoluble formazan precipitate that can be quantified spectrophotometrically. For the MTT assay, the medium was carefully removed, and the scaffolds were transferred to a clean 96-well plate. A 200 µL volume of MTT solution (50 µg/mL final concentration) was added to each scaffold, and the plate was incubated at 37 °C for 4 h in 5% *v*/*v* CO_2_. After incubation, the medium was removed and replaced with 100 μL of isopropanol for blue formazan crystal solubilization. Finally, the absorbance values from the supernatant were recorded at 570 nm by a Multiskan Fc Microplate Photometer (Thermo Fisher Scientific, Waltham, MA, US), and the cellular viability percentage of the cells grown onto the Cs/n-HAp scaffolds was calculated relative to the cellular viability of the cells grown onto the pure Cs scaffold.

#### 2.5.3. RNA Isolation, Reverse Transcription, and Real-Time PCR for Gene Expression Analysis

The response of NIH-3T3 cells grown on the pure Cs and Cs/n-HAp scaffolds was also evaluated in terms of mRNA expression levels of the mouse *Casp 3* gene using a real-time PCR assay. Total RNA and protein extraction from cells grown on scaffolds for 72 h was performed using the AllPrep DNA/mRNA/protein mini kit (Qiagen—Milano, Italy). Briefly, the scaffolds were incubated in a suitable volume of RLT Lysis Buffer (with β-mercaptoethanol added, according to the manufacturer’s instructions), and then opportunely homogenized by a mini-potter. mRNA and protein extraction was carried out according to the kit’s protocol. At the end of the extraction protocol, RNA aliquots were kept stored in RNase-free conditions at −80 °C until use. RNA concentrations were quantified by spectrophotometry and λ_260_/λ_280_ ratios were calculated to evaluate possible protein contamination.

In order to investigate the mRNA expression of cells seeded on different types of scaffolds, cDNA was prepared by reverse transcription of 100 ng RNA in the presence of random primers using the Bio-Rad iScript Select cDNA Synthesis kit (Bio-Rad, Milan, Italy), according to the manufacturer’s instructions. The reference sequence of the Casp3 mRNA was collected from the NCBI gene database (https://www.ncbi.nlm.nih.gov/gene/ accessed on 10 July 2024), and analyzed to identify suitable oligonucleotide tracts to be used as primer pairs in the real-time PCR (qPCR) assays. Forward and reverse primers were designed in adjacent exons. AmplifX version 1.5.4 software was used (http://ifrjr.nord.univ-mrs.fr/AmplifX accessed on 10 July 2024) to validate PCR size, GC content, end stability, and self/cross-dimer formation of the selected oligonucleotide sequences, which were finally purchased from Eurofins Genomics (Germany). In the following Table 2, primer sequence details are reported. Before qPCR analysis, each primer pair was tested for efficiency according to the ‘amplification efficiency parameters for genes of interest and internal controls’ proposed by Schmittgen and Livak [36]. qPCR was performed using the iQ SYBR Green Supermix protocol (Bio-Rad) with a Rotor-Gene 3000 (Corbett Research, St. Neots, UK) real-time thermal cycler. The 28S ribosomal RNA was used as the qPCR internal control to normalize mRNA expression. The qPCR output data (i.e., threshold cycle value, C_t_) were represented as 2^−ΔCt^ values proportional to the amount of the detected target mRNA (ΔC_t_ = target gene C_t_ − standard gene C_t_) [36]. Mean values were calculated from two different rounds of qPCR for three biological replicates; in graphics, results are reported as expression level fold-change vs. the control condition (control fold-change = 1).

### 2.6. Statistical Analysis

Results are representative of three independent experiments, where at least triplicate specimens were tested, unless differently stated. Data are expressed as the mean ± standard deviation of measurements. Statistical analysis was performed using the Student’s *t*-test and *p* < 0.05 was considered as statistically significant.

## 3. Results

### 3.1. n-HAp Nucleation into Chitosan Matrixes to form Cs/n-HAp Composite: Physicochemical Analysis

Cs/n-HAp composites with different weight percentages of inorganic phase were obtained by growing crystals into the scaffold structure, as assessed by comparing the composite XRD patterns with that of the reference HA powder (ICDD PDF-4 database, card 00-055-0592) (Figure 1B). The mineral contents in the nanocomposite scaffolds were verified trough the TGA data using Equation (4). From this calculation, we found that there was a close match between the experimental amounts of n-HAp in the composite scaffolds and the theoretical n-HAp amounts based on the reactants introduced into the slurry. HA nanocrystals successfully formed in the scaffold structure by converting their precursors into n-HAp, thanks to the pH increasing due to the presence of L-arginine in the developed media. It can be hypothesized that in situ biomineralization began with the formation of amorphous CaP (ACP) from the precursor salts. This compound has high reactivity and can therefore transform to apatite in several ways. Being that the scaffolds were immersed in the basic solution with L-arginine, ACP, which is extremely unstable, ripened into the most thermodynamically stable phase [37], giving rise to crystalline HA resembling the deproteinated bone apatite phase both in crystallinity and nanoscale size [38]. Figure 1C displays the XRD patterns of Cs/Ca-P precursor 30 (which was about to become scaffold Cs/n-HAp 30%) at three different times (5 min, 6 h, and 24 h) during the nucleation process. Characteristic diffraction peaks of HA (i.e., 002 and 211) were evident at both 6 and 24 h, while they were almost absent in the sample in which the reaction was interrupted after 5 min. The primary domain size along the c-axis, calculated by Scherrer’s formula for the (002) reflection peak (2θ ≈ 26°) of the XRD pattern at 6 h, was about 10 ± 2 nm, while for the XRD pattern at 24 h, the domain size was about 13 ± 2. It can be hypothesized that, during the first reaction phase, ACP was stabilized by the three-dimensional matrix of chitosan and converted to crystalline HA in the following schedule. The growth of crystals from the precursor phase with an amorphous structure, in fact, could be altered by the presence of additives/impurities or simply by the presence of a polymeric matrix within the crystallizing medium [39]. However, when the polymeric matrix was immersed in an aqueous solution with basic pH, the hydroxyl ions (OH^−^) could penetrate the structure and provoke the conversion of ACP to nanocrystalline HA, according to the following equation [40]:(5)$${PO}_{4}^{3-}+ACP+{OH}^{-}\to {Ca}_{10}{{(PO}_{4})}_{6}{\left(OH\right)}_{2}$$

The representative macrostructures of pure Cs crosslinked with arginine and Cs/n-HAp composites are presented in Figure 2A,B, showing the scaffold’s homogeneous and isotropic pore distribution, with a circular or elliptical shape. It is worth noting that porosity homogeneity was improved through a series of parameters and process strategy optimization (i.e., acetic acid solvent removal after freeze-drying, arginine concentration lowering, and so on). This fine-tuning, extensively described in Appendix A, allowed us to enhance some of the performance reported for scaffolds obtained in our previous works [33,41].

Because of these improvements, the arginine-mediated biomineralization optimized in this paper, while it allowed n-HA crystal growth into the scaffold, did not strongly affect the scaffold microstructure. The pore diameter of the arginine-stabilized chitosan and composite scaffolds ranged from 90 to 170 μm and did not seem to be affected by the nano-crystal percentage. Image analysis with ImageJ 1.53t software confirmed that the pore size was quite heterogeneous, but the pore size with the major frequency was in the range 95–125 μm (Figure 2B), which slightly increased up to being centred around 140 μm for the Cs/n-HAp 30 scaffold. A high porosity percentage superior to 95% for each scaffold typology (Table 3) was evaluated through Equation (1). However, a slight reduction in pore diameter of the chitosan scaffold treated with arginine with respect to the untreated one could be detected (data not shown, please see ref. [34]) and ascribed to the formation of a chemical bond between chitosan and arginine. The above hypothesized interactions occurring through hydrogen bonds allowed the approach of chitosan molecules and, consequently, the pore diameter decreased. This hypothesis was consistent with the diameter and thickness reduction of the arginine-stabilized scaffold, which became about 15% lower with respect to the pure Cs sample (data not shown, please see ref. [34]). n-HAp uniform distribution into the composite structure was evaluated by EDS mapping, which highlighted a homogeneous amount of calcium phosphate on the scaffold trabeculae surface (Figure 2C). In addition, it was possible to quantify the atomic percentages of calcium and phosphorous, which allowed estimation of a Ca/P ratio of about 1.9. Moreover, the non-normalized concentration in weight percent of the elements (Table 3) allowed confirmation of the direct proportionality between the calcium and phosphorous precursors inserted into the matrix and the formed n-HAp.

FTIR spectroscopy was applied on both the composite scaffolds and pure Cs stabilized with arginine to unravel the arginine–chitosan chain interaction. The spectrum of the chitosan membrane stabilized with arginine (Figure 3A) showed the characteristic band of Cs at 1650 cm^−1^ and the band of arginine at 1419 cm^−1^. The new band at 1585 cm^−1^ was most likely the result of a convolution of the contemporary shift of the bands at 1630 cm^−1^ and 1528 cm^−1^ in the Cs membrane and arginine spectra, respectively. The interaction provoked an elongation of C=N bonding in the guanidine group, which in turn generated a transmittance band at lower frequency. Furthermore, there was a small shift from 1160 cm^−1^ to 1154 cm^−1^ in the C−O−C asymmetric stretching of chitosan, which confirmed this hypothesis. There were further shifts in the C−O stretching vibrations of the pyranose ring from 1094 cm^−1^ to 1087 cm^−1^ and from 1045 cm^−1^ to 1057 cm^−1^. Therefore, we could hypothesize that the linkage occurred, on the one hand, between the positively charged amino group of one Cs chain and the α-carboxylate group of arginine, and on the other hand, between the guanidine moiety of arginine and the hydroxyl group of a second Cs chain (Figure 3B).

### 3.2. Swelling Behavior

The swelling ratio profiles of both the pure Cs and Cs/n-HAp scaffolds are shown in Figure 3C. Cs-based freeze-dried scaffolds are known for their high percentage of swelling. The pure chitosan scaffold showed a higher swelling ratio compared with the composite scaffolds in which the swelling percentage was reduced [24]. The graph shows that the swelling ratio (Q) at 72 h was about 48% for pure chitosan and was between 20% and 30% for all types of composite scaffolds. Moreover, the Q decrease seemed to be function of the n-HAp amount in the scaffold. This significant difference between the pure and composite scaffolds might have been due to the binding of HA to the chitosan hydrophilic groups (NH_2_ and OH), leading to a lowering of the water content. Moreover, a slight lowering of scaffold elasticity could be induced by the nucleation of inorganic particles into the matrix.

### 3.3. Mechanical Properties

The mechanical properties of the chitosan and composite scaffolds are summarized in Table 4. The Young’s modulus at low strain values (E_low_) of the scaffolds increased from 2.1 ± 0.2 kPa for pure chitosan to 8.35 ± 0.55 kPa for Cs/n-HAp 30 wt%, while the slope values of composites at 20 and 30 wt% n-HAp did not significantly differ. This result agreed with the swelling ratio evaluation, showing that there was no significant difference between composites at 20 and 30 wt% n-HAp. Although no significant differences could be detected among the slope values of composites at 20 and 30 wt% n-HAp, the stiffness of the n-Hap-loaded scaffolds showed a slight increase, indicating that the in situ biomineralization method enhanced the mechanical properties of the composite scaffolds. Likewise, the Young’s modulus at high strain values (E_high_) increased as the HAp % increased, passing from 30.26 ± 1.77 kPa for pure chitosan to 81.94 ± 5.16 kPa for Cs/n-HAp 30 wt%, while the strain value at which the stress–strain curve slope abruptly changed did not significantly differ. We hypothesize that, in the described process, amounts of n-HAp greater than 20 wt% did not grow inside the chitosan trabeculae anymore but started to cover the pore surfaces without conferring any increase in mechanical strength at low strain values, which, instead, was more evident at higher strain values when the scaffold structure was almost totally collapsed due to compression.

### 3.4. Viability Assay

The basic cytocompatibility of NIH-3T3 cells seeded onto the Cs and Cs/n-HAp scaffolds was assessed by the MTT viability assay after 48 h of culture, in accordance with the literature [42,43,44,45]. The viability of fibroblast-like cells (expressed as cell viability percentage normalized to the cells grown on pure Cs as the control) after 48 h is shown in Figure 4A. The Cs/n-HAp scaffolds with a lower percentage of n-HAp (Cs/n-HAp 10) showed cell proliferation almost like the control Cs, suggesting that this n-HAp concentration did not induce a significant difference in cell proliferation. Conversely, for both the scaffolds containing higher concentrations of HA (i.e., Cs/n-HAp 20 and Cs/n-HAp 30), decreases in cell proliferation were detected (down to 62% and 56% proliferation, respectively). Interestingly, this evidence may be related to the swelling ratios of Cs/n-HAp 20 and Cs/n-HAp 30, which were both lower in comparison to the pure chitosan scaffold and Cs/n-HAp 10 scaffold (Figure 3C). In addition, it must be noticed that their mechanical properties (Table 4) were also significantly greater than the Cs as well as the Cs/n-HAp 10 scaffolds. Intriguingly, both of these structure-related factors were inversely correlated with the proliferation data; most likely, swelling and stiffness may influence cell proliferation by affecting the ability of cells to colonize and/or penetrate the polymeric matrix.

### 3.5. Quantitative Real-Time Polymerase Chain Reaction (PCR)

By extracting RNA from NIH-3T3 cells grown for 72 h onto the Cs and Cs/n-HAp scaffolds, Casp-3 mRNA expression analysis was performed. Remarkably, while low basal levels of Casp3 mRNA expression were detected in Cs, Cs/n-HAp 10, and Cs/n-HAp 20 (without significant difference), in NIH-3T3 cells grown on the Cs/n-HAp 30 scaffold, strong transcriptional activation of Casp-3 expression occurred (Figure 4B). In particular, a ~512 positive fold-change in Casp-3 mRNA levels was detected with respect to the other scaffolds. Thus, it could be noticed that, according to the previous considerations for proliferation, the decrease in cell proliferation found in Cs/n-HAp 20% may have majorly depended on a lower number of penetrated and adhered cells in the early phases post seeding; on the other hand, the reduced proliferation of cells grown on the 30% n-HAp-loaded scaffold might have been related to induction of the apoptotic cell death pathway, which did not occur in cells on the other scaffolds.

### 3.6. Multilayer Assembly

The multilayer graded structure was obtained by assembling pure Cs and Cs/precursors loaded on a frozen scaffold, as reported in the Materials and Methods section. Briefly, each gelatinous suspension (Cs, Cs/n-HAp precursor) was poured into properly designed ring-shaped PLA molds (Figure 5). These layers, frozen separately, were stacked as described in Figure 5, with the help of another cylindrical PLA mold, and glued to each other by a suitable volume of chitosan suspension. The successive solvent removal and n-HAp nucleation phases were executed in the same manner as for the single-layer scaffolds. The adhesion of the individual layers was morphologically verified through SEM analysis after 6 months in Tris-HCL medium. In Figure 5, the transverse section of the multilayer structure after immersion in tris-HCL and freeze-drying is shown, highlighting the contact surfaces between layers with different nanohydroxyapatite (n-HAp) contents. As proof-of-concept, in Figure 5, we report the stacking of three different layers up to 20 wt% of n-HAp (i.e., 0%, 10%, 20%), although this method allows for the stacking of multiple chitosan layers. The different percentages of the elements constituting the multilayer structure were verified by EDX quantification. The Ca/P elements ratio was verified to be constant, even if the percentage increased proportionally, with the same trend verified for separate layers.

## 4. Discussion

This paper aimed to describe the design, fabrication, and characterization of a multilayered-tissue-mimicking structure. In particular, the osteochondral unit was taken as an example because of the urgency of designing structures useful for mimicking this tissue. Osteochondral lesions, in fact, are recognized to affect both the articular cartilage and the transition from cartilage to bone, specifically at the cartilage–bone interface. Despite its significance, the tidemark has remained largely unexplored in over 90% of the research published in the past decade [46].

The native osteochondral interface can be considered either as a stratified or graded element. The cartilage-to-bone interface is hierarchically divided into distinct yet continuous zones of uncalcified cartilage, calcified cartilage, and subchondral bone. Therefore, in this study, we aimed to follow an approach that mimicked the interface as a continuous graded structure composed of layers of chitosan (cartilage-like) with increasing percentage of hydroxyapatite (bone-like).

The synthesis of Cs/n-HAp composites, achieved through crystal growth within the scaffold structure, was demonstrated to be a viable method for creating materials with adjusted inorganic phase percentages. The confirmation of mineral content in the nanocomposite scaffolds through TGA data established the reliability and accuracy of the synthesis process, aligning well with theoretical expectations. The pH-induced biomineralization process, facilitated by the presence of L-arginine, played a crucial role in converting precursors into amorphous CaP (ACP) and, subsequently, crystalline HA. The dynamic evolution of crystal growth, as evidenced by XRD patterns at different time points, provided insight into the temporal progression of the transformation from ACP to crystalline HA. The calculated increase in primary crystal size along the c-axis over time highlighted the transformation from ACP to crystalline HA, indicating a controlled and dynamic process. The SEM analysis highlighted the effectiveness of the synthesis process in achieving homogeneous and isotropic pore distribution in both the pure Cs and Cs/n-HAp composite scaffolds. The minimal impact of the arginine-mediated biomineralization process on scaffold microstructure, with a slight reduction in pore diameter, suggested a well-controlled and optimized synthesis. The successful incorporation of n-HAp was evidenced by the uniform distribution of calcium phosphate on the scaffold trabeculae surfaces, as confirmed by EDS mapping.

The investigation into the swelling characteristics and mechanical properties of the chitosan and composite scaffolds offered valuable insight into their behaviors and potential applications. Pure chitosan exhibited a higher swelling ratio compared to the composite scaffolds, and this difference becomes particularly pronounced at 72 h. The observed reduction in swelling percentage in the composite scaffolds, especially in the presence of 30% n-HAp, suggested an intricate interplay between hydroxyapatite (HA) binding to chitosan’s hydrophilic groups (NH_2_ and OH) and the resulting decrease in water content.

Biomineralization led to a 75% increase in the elastic modulus of chitosan. Interestingly, the lack of significant difference in slope values between the composites at 20 and 30 wt% of n-HAp implied that higher n-HAp amounts may not lead to a proportional increase in mechanical strength but instead result in the coverage of pore surfaces.

The comprehensive assessment of cytocompatibility and cellular response to the chitosan (Cs) and chitosan/hydroxyapatite (Cs/n-HAp) scaffolds yielded valuable insight into the intricate interplay between biomaterial properties and cellular behavior.

In examining the cytocompatibility of the scaffolds using the MTT viability assay, it became evident that the Cs/n-HAp 10 scaffold exhibited cell proliferation comparable to the control Cs, suggesting good biocompatibility at this n-HAp concentration. However, a contrasting trend was observed with the Cs/n-HAp 20 and Cs/n-HAp 30 scaffolds, wherein decreases in cell proliferation were noted, reaching up to 62% and 56%, respectively. This concentration-dependent effect was intriguingly associated with the swelling ratio and mechanical properties of the scaffolds. The inverse correlation between swelling, stiffness, and cell proliferation suggests a complex interplay, whereby these structural factors may influence the ability of cells to interact with and penetrate the polymeric matrix.

Moving to the molecular level, the quantitative real-time PCR analysis of Casp-3 mRNA expression provided further insight. Cs, Cs/n-HAp 10, and -Cs/n-HAp 20 exhibited low basal levels of Casp-3 expression without significant differences. By contrast, the Cs/n-HAp 30 scaffold showed remarkable and significant transcriptional activation of Casp-3 expression, indicating a potential induction of the apoptotic cell death pathway. This observation aligned with the reduced cell proliferation in Cs/n-HAp 30%, suggesting a link between the concentration of hydroxyapatite and the activation of apoptotic processes.

The fabrication of a multilayer graded structure represents a significant advancement in scaffold design, combining pure Cs and Cs/precursor-loaded scaffolds. This innovative approach involves the pouring of gelatinous suspensions (Cs, Cs/n-HAp precursor) into a custom-designed PLA mold, followed by stacking of separately frozen layers.

The stability of the adhesion between layers with varying n-HAp content, as evidenced by observations of the transverse section of the multilayer structure after a 6-month stability test, validated the successful assembly method pointed out in this study. The compositional stability observed in the multilayer structure opens new possibilities for tailoring material properties across different layers, paving the way for enhanced functionality and performance in regenerative medicine applications.

## 5. Conclusions

With the aim of joining the possibility of tailoring scaffold porosity and shape through the freeze-drying technique with the opportunity of growing a bone-like mineral phase inside the polymer, a graded composite Cs-nucleated HAp scaffold was developed that mimics the biochemical and biophysical properties of the different layers of native osteochondral structures. The innovative method proposed here, which combines the nucleation of bone-like n-HAp into a Cs matrix with an original stacking technique, allowed for the design of a multilayer composite scaffold at increasing concentrations of n-HAp along the z-axis, as a potential osteochondral substitute. The configuration of such a graded composite Cs-nucleated HAp scaffold can be easily tailored as a function of the required structure and presents proof-of-concept that multilayered tissues can be mimicked.

Despite this, it is obvious that, in the future, other aspects such as the influence of the cellular environment (chondrocytes, osteoblasts, etc.) and the use of biomolecules will need to be considered in addition to composition to achieve mechanical properties comparable to those of the native interface and capable of inducing vascularization.

## Figures and Tables

**Figure 1 bioengineering-11-00471-f001:**
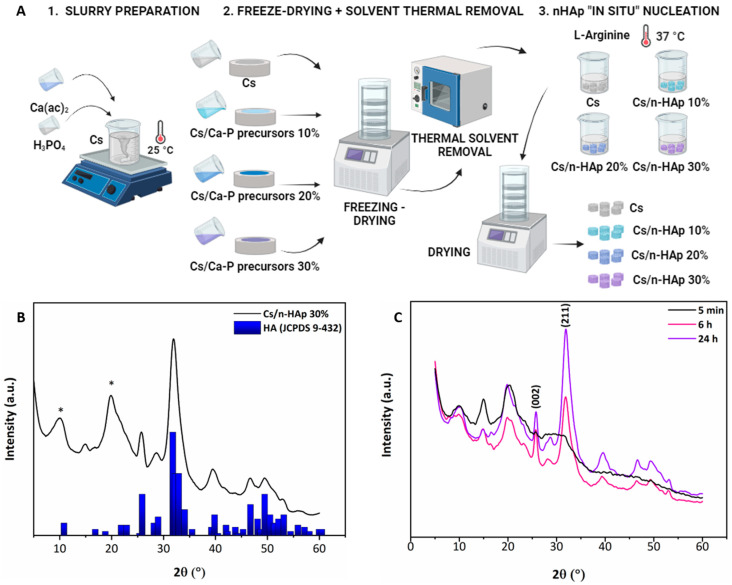
(**A**) Scheme of Cs/n-HAp composite scaffold preparation. (**B**) XRD pattern of Cs/n-HAp 30% compared with reference HA powder (JCPDS 9-432). Asterisk (*) refers to Cs diffraction maxima. Since Cs is a semi-crystalline material, a broad diffraction peak at about 2θ ≈ 20° was detected. (**C**) XRD patterns of Cs/n-HAp 30% obtained from Cs/Ca-P precursor 30 at three different soaking times (5 min, 6 h, and 24 h). The Miller indices of two diffraction peaks characteristic of HA are shown (002), (211). Asterisk (*) refers to Cs diffraction maxima.

**Figure 2 bioengineering-11-00471-f002:**
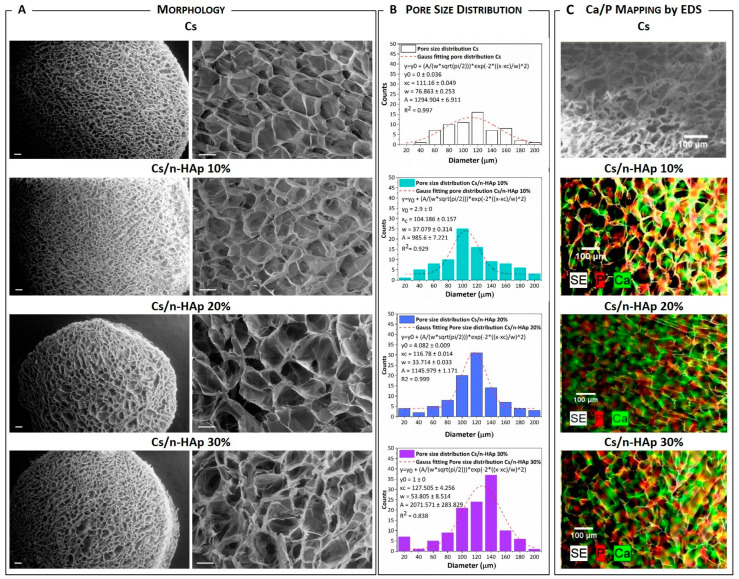
(**A**) SEM micrographs of pure Cs and Cs/n-Hap composite scaffolds’ transverse sections [left side: mag. 60×, scale bar 200 µm; right side: mag. 300×, scale bar 100 µm). (**B**) Pore size distribution analysis of pure Cs and Cs/n-Hap composite scaffolds. (**C**) Calcium (Ca) and phosphorus (P) mapping by EDS analysis of pure Cs and Cs/n-Hap composite scaffolds’ transverse sections (Mag. 150×; scale bar 100 μm).

**Figure 3 bioengineering-11-00471-f003:**
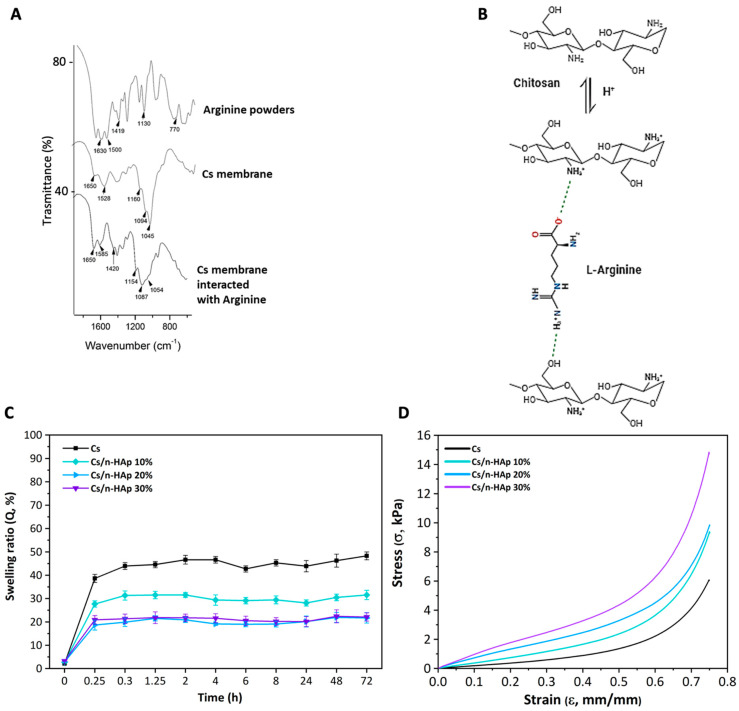
(**A**) FTIR spectra of L-arginine powder, pure Cs membrane, and Cs membrane interacted with L-arginine. (**B**) Representation of the chemical interaction between chitosan and L-arginine. (**C**) Swelling ratio (expressed as percentage) of pure Cs scaffold and composites Cs/n-HAp 10%, Cs/n-HAp 20%, and Cs/n-HAp 30%. (**D**) Stress–strain compression curves of pure Cs scaffold and composites Cs/n-HAp 10%, Cs/n-HAp 20%, and Cs/n-HAp 30%.

**Figure 4 bioengineering-11-00471-f004:**
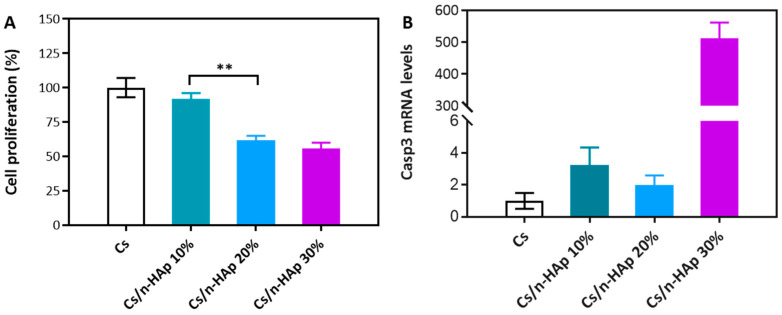
(**A**) MTT proliferation assay of NIH-3T3 cells cultured on pure Cs scaffold and composite scaffolds (Cs/n-HAp 10%, Cs/n-HAp 20%, and Cs/n-HAp 30%) after 48 h of cell seeding. Data are expressed as mean percentage ± SEM (*n* = 3). The cell proliferation on the pure Cs scaffold has been normalized to 100% (control). Statistical analysis was conducted with the Student’s *t*-test (** *p* <0.01), and the difference between the other groups is not significant. (**B**) Transcriptional regulation of Casp-3 gene in NIH-3T3 cells grown for 72 h on Cs and composite scaffolds. Quantitative expression analysis by real-time RT-PCR. Expression of the identified target mRNA is represented as the 2^−ΔCt^ function value, obtained from amplification data (Ct, threshold cycle), normalized by 28S RNA values (internal control for calibration). The 2^−ΔCt^ values are relative to the control mean value (Cs = 1) and are expressed as fold-change on the y-axis in the graphic. Mean values (±S.E.M.) are derived from *n* = 3 independent assays.

**Figure 5 bioengineering-11-00471-f005:**
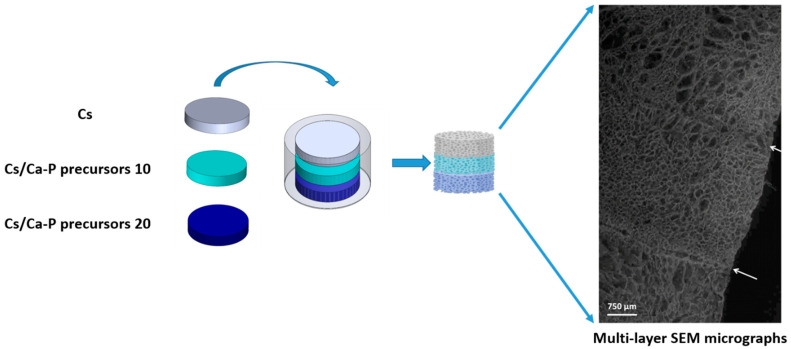
Scheme for the Cs and Cs/precursor assembly mode and SEM image of the obtained multilayer graded structure. Arrows highlight the border line between layers with increasing n-HAp loading.

**Table 1 bioengineering-11-00471-t001:** Cs and Cs/precursor compositions.

Scaffold	Acetic Acid (% *w*/*v*)	Cs	Ca(ac)_2_ (mg/mL)	H_3_PO_4_ (mg/mL)
Cs	1	1.67	-	-
Cs/Ca-P precursor 10	1		23	8.9
Cs/Ca-P precursor 20	2		46	17.85
Cs/Ca-P precursor 30	3		69	26.775

**Table 2 bioengineering-11-00471-t002:** Details of primer sequences for the qPCR assays. For each gene, the NCBI accession number of the mRNA reference sequence (Ref Seq mRNA; rDNA for the 28S ribosomal gene) used for primer design is reported. Gene-specific nucleotide sequences are reported as forward (sense) and reverse (antisense) primers in the 50–30 direction. For each primer pair, the expected amplicon length is reported (PCR product size) as base pairs (bp).

Gene	Ref Seq mRNA acc. n.	5′-3′ Sense	5′-3′ Antisense	PCR Size (bp)
Casp-3	NM_001284409.1	CTGGACTGTGGCATTGAGAC	TAACCAGGTGCTGTAGAGTA	157
28S	NR_003279.1	CGTGAGACAGGTTAGTTTTAC	ATCCCACAGATGGTAGCTTC	143

**Table 3 bioengineering-11-00471-t003:** Mean diameter from SEM micrograph analysis by ImageJ software, porosity by Equation (1), and calcium (Ca) and phosphorus (P) quantification by EDS (unn. C. = un-normalized concentration in weight percent of the element).

Scaffold	Mean Diameter (µm)	Porosity (%)	Unn. C. (wt%)	
			Ca	P
Cs	111 ± 1	97.88 ± 0.09	------	------
Cs/n-HAp 10	104 ± 1	97.80 ± 0.12	7.61 ± 1.29	2.95 ± 0.64
Cs/n-HAp 20	117 ± 1	97.54 ± 0.04	13.49 ± 1.26	5.12 ± 0.80
Cs/n-HAp 30	128 ± 4	98.10 ± 0.08	20.02 ± 0.95	7.86 ± 0.76

**Table 4 bioengineering-11-00471-t004:** Mechanical properties of Cs/n-HAp composite scaffolds: E_low_ at low strain values, E_high_ at high strain values, and ε_y = 0._

Scaffold	E_low_ (kPa)	E_high_ (kPa)	ε_y_
Cs	2.16 ± 0,16	30.26 ± 1.77	0.581 ± 0.006
Cs/n-HAp 10	3.49 ± 0.13	55.55 ± 2.88	0.590 ± 0.001
Cs/n-HAp 20	7.86 ± 0.41	67.78 ± 3.20	0.584 ± 0.007
Cs/n-HAp 30	8.26 ± 0.50	81.94 ± 5.16	0.577 ± 0.005

## Data Availability

Data are contained within the article.

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
