# Peer review of "Towards Complex Tissues Replication: Multilayer Scaffold Integrating Biomimetic Nanohydroxyapatite/Chitosan Composites"

_bioengineering, 2024, doi:10.3390/bioengineering11050471_

Round 1
Reviewer 1 Report
Comments and Suggestions for Authors
Based on the provided text, the claim of the research being osteochondral seems unsupported by the data presented. Here are some critical points to consider:
1. The study primarily focuses on the fabrication and characterization of chitosan/nano-hydroxyapatite (Cs/n-HAp) composite scaffolds. While these scaffolds may have potential applications in tissue engineering, particularly in bone regeneration due to the presence of hydroxyapatite, there's no explicit evidence or discussion about their suitability for osteochondral applications.
2. The text emphasizes physico-chemical analysis, including XRD patterns, TGA data, FTIR spectroscopy, swelling behavior, and mechanical properties. These are essential for understanding the material properties but do not directly correlate with osteochondral applications.
3. Viability assays (MTT assay) and gene expression analysis (Casp-3 mRNA expression) were conducted using NIH-3T3 cells cultured on the scaffolds. While these assays provide insights into cytocompatibility and potential cellular responses, they do not assess osteochondral tissue formation or regeneration.
4. The decrease in cell proliferation and the activation of Casp-3 expression on scaffolds with higher nano-hydroxyapatite content might suggest adverse effects on cell behavior, but it does not establish the suitability of the scaffolds for osteochondral applications.
5. The multilayer graded structure is described, but its relevance to osteochondral applications is not clearly addressed. While the assembly method is detailed, and the presence of nano-hydroxyapatite in different layers is mentioned, there's no indication of how this structure mimics the complex interface between bone and cartilage in osteochondral tissue.
6. The text does not mention any in vivo studies or animal models to evaluate the scaffolds' performance in promoting osteochondral regeneration.
7. In vivo studies are crucial for assessing the efficacy and safety of tissue-engineered constructs, especially for complex tissues like osteochondral tissue.
Comments on the Quality of English Language
Fine
Reviewer 2 Report
Comments and Suggestions for Authors
Great work!
Author Response
We thank the reviewer for appreciating our work.
Reviewer 3 Report
Comments and Suggestions for Authors
The manuscript reports a regenerative approach for osteochondral lesions. The use of chitosan matrices is not new but the work is well conducted and supported by the results. The Introduction and References sections covers most of the relevant literature data.
Comments on the Quality of English LanguageNo comments
Author Response

(The authors gave the same response as above.)

Reviewer 4 Report
Comments and Suggestions for Authors
It is a well written and interesting manuscript. Author names in reference 41 must be written according to journal style, that is avoiding capital letters where are not appropriate.
Author Response
We thank the reviewer for appreciating our work. We have modified reference 41 as suggested, which has now become reference 43 after the revisions.
Round 2
Reviewer 1 Report
Comments and Suggestions for Authors
The title of the manuscript is misleading. This manuscript has no relevance to osteochondral lesion repair. The author reporting a false claim.
The chitosan/nanohydroxyapatite (Cs/n-HAp) scaffolds are checked for biocompatibility only. No data is reported to support their claim for osteochondral lesion repair.
The manuscript could not be accepted in this form.
Comments on the Quality of English Languagenil
Author Response
Point-by-point response to Reviewer 1
- The title of the manuscript is misleading. This manuscript has no relevance to osteochondral lesion repair. The author reporting a false claim. The chitosan/nanohydroxyapatite (Cs/n-HAp) scaffolds are checked for biocompatibility only. No data is reported to support their claim for osteochondral lesion repair. The manuscript could not be accepted in this form.
We appreciate the reviewer's suggestion, which prompted us to reconsider the potentially misleading title. The aim of this study, as stated in the text, is to mimic the structure of osteochondral tissue: "Following this logic, the design of a graded composite Cs-nucleated HAp scaffold, which mimics the biochemical and biophysical properties of the different layers of native osteochondral structures is proposed." Accordingly, we have modified the title as follows: "Hybrid nanohydroxyapatite/chitosan composites as multilayer scaffold mimicking the osteochondral tissue”.
Additionally, in response to the reviewer's feedback, we have corrected the last sentence in the abstract, substituting "regenerating massive" with "mimicking" in the abstract.
We hope these revisions address the reviewer's concerns adequately.
Furthermore, please find attached the revised manuscript reflecting the changes made in response to the reviewer's suggestions.

Round 3
Reviewer 1 Report
Comments and Suggestions for Authors
The title is "Hybrid nanohydroxyapatite/chitosan composites as multilayer 5 scaffold mimicking the osteochondral tissue" to my surprise no osteocytes or chondrocytes are checked.
No osteogenic or chondrogenic differentiation/compatibility/ maturation is evaluated on the scaffold.
Alternatively, no stem cell differentiation is performed on the scaffold.
Only NIH 3T3 cells were checked by MTT assay and the caspase 3 gene was checked, which is not sufficient to conclude osteochondral regeneration or compatibility.
Additional Cell culture based assays are required to be performed to this manuscript published.
The presented data is inadequate to support the use of the scaffold in biological applications.
The manuscript in current form cannot be accepted for publication.
Author Response
We are grateful for the reviewer suggestion and have worked diligently to make the necessary changes to our work. We have completely revised the title of the manuscript to better reflect its content and have adjusted the introduction to align with the new title.Thank you again for your guidance in reviewing our work.